# MODALITY-BALANCED DECOUPLING ALIGNMENT FOR TEXT-VIDEO RETRIEVAL

## ABSTRACT

Text-video retrieval, the task of retrieving videos given a text query or vice versa, plays a significant role in video understanding. A significant challenge in this task is the semantic gap between video and text, primarily caused by the disparity in information capacity and the highly coupled nature of video information. Existing alignment methods mainly focus on multi-grained alignment between videos and text, which fails to address the capacity imbalance between video and text feature space. To address these issues, we propose Modality-Balanced Decoupling Alignment (MBDA) , a novel method that align the two modalities with closer distribution and more balanced information capacity in the feature space. Specifically, our model consists of two modules. The Modality Proximity Alignment module brings the video embedding closer to the text embedding, while the Video Representation Orthogonal Decoupling module separates the aligned video embedding into two orthogonal components, achieving better balance with their textual counterparts. Furthermore, we demonstrate that our decoupling approach achieves orthogonality while eliminating information redundancy among components through low-rank decomposition and frequency-domain analysis via Discrete Fourier Transform. The proposed method improves the baseline by a large margin. Extensive experiments demonstrate that MBDA achieves state-of-the-art performance on four most widely used public benchmarks, MSR-VTT(52.4%), DiDeMo(53.1%), MSVD(54.0%), and ActivityNet(49.6%).

## 1 INTRODUCTION

With the explosive growth of video content on the internet, tasks related to video understanding have become increasingly significant. Among them, Text-video retrieval (Yu et al., 2017), which aims to retrieve relevant video clips based on textual queries or vice versa, has gained widespread applications and thus attracts extensive exploration these days. However, the task remains inherently challenging due to the modality gap between text and video. As Figure 1(a) shows, this difficulty is further exacerbated by the semantic imbalance between the two modalities: videos often contain rich, fine-grained information, while their associated textual descriptions, such as captions, subtitles, or summaries tend to be sparse and coarse.

To further deal with this challenge, modality alignment is typically applied after obtaining video and text representations from the multimodal encoders. These efforts have evolved from global matching approaches, such as video-sentence alignment (Liu et al., 2022; Gabeur et al., 2020), to more fine-grained strategies, including frame-word alignment (Wang et al., 2022) and multi-level matching (Ma et al., 2022). However, these methods still fall short of fundamentally bridging the modality gap because of the capacity imbalance between video and text is not solved.

As shown in Figure 1(b), when visualizing the shared feature space using t-SNE (Maaten, 2014), we observe a significant imbalance in the semantic volumes occupied by video and text features, which poses a major challenge for direct alignment. Methods like T-MASS (Wang et al., 2024) try to alleviate this imbalance by expanding the text feature space. However, this method determines the optimal text-video match through a search process, which results in significant computational overhead, leading to a much prolonged inference phase. Differently, we propose to address the issue by decoupling video features into components with smaller capacity of feature space. However, there is another challenge: the information encoded in video features is highly entangled across multiple

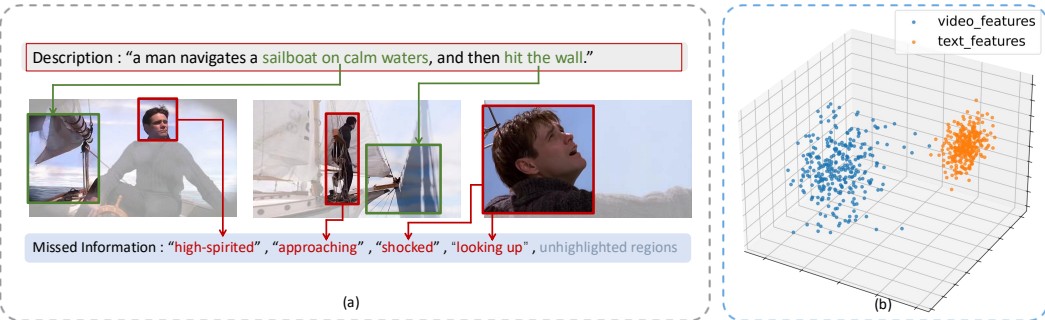

Figure 1: There is a significant information gap between video and text. (a) shows that textual descriptions often fail to fully capture all the information contained in a video. (b) shows that we visualize the distribution of a set of video and corresponding text representations in the embedding space, which reveals that the video representation space is significantly larger than that of the text.

dimensions such as spatial and temporal dimensions, making it inherently difficult to establish a direct correspondence between video and text representations.

To address the challenges above, we propose a novel matching method termed **Modality-Balanced Decoupling Alignment**, which explicitly balances the information capacity of video and text embeddings during the alignment process. Specifically, to facilitate the alignment of decoupled video representations with text, we first perform a Modality Proximity Adjustment (MPA) step to reduce the distance between video and text embeddings in the representation space. Next, we introduce an orthogonal vector decoupling technique named Video Representation Orthogonal Decoupling (VROD). In this step, the video representation is decoupled into two orthogonal vectors by downsampling on different dimensions, each capturing different aspects of the original embedding. Orthogonality achieves balanced modality alignment by transforming the high-capacity video representation into lower-capacity components, each of which is more comparable in scale to the text representation, thus enabling a more precise and balanced alignment. To prove the orthogonality of the VROD decoupling, subsequently, each vector is further factorized into two low-rank components, which are then transformed into the frequency domain using the Discrete Fourier Transform (DFT) (Cooley & Tukey, 1965). In this domain, we prove that the resulting components are mutually orthogonal, indicating that the video information is effectively decoupled with minimal redundancy. This leads to a significantly reduced semantic space for the video representation, which becomes comparable in scale to that of the text.

We theoretically derive and visually demonstrate: (1) the MPA step effectively reduces the modality gap by aligning video representations closer to the textual space, (2) our decoupling method successfully produces orthogonal representations, resulting in a more balanced representation space size between video and text. Then we evaluate our method on four benchmark datasets, i.e., MSR-VTT (Xu et al., 2016), DiDeMo (Anne Hendricks et al., 2017), ActivityNet (Caba Heilbron et al., 2015), and MSVD (Chen & Dolan, 2011). Our approach achieves state-of-the-art performance on all of these datasets, showing the advantages of the proposed method.

The main contributions are as follows: (1) We provide an insightful and intuitive analysis of the fundamental limitation in text-video retrieval: the imbalance of information capacity between modalities and the highly coupled nature of video information, which are major obstacles to effective cross-modal alignment. (2) To alleviate such limitation, we propose a novel video representation decoupling method that pulls video embeddings closer to text embeddings and decouple them into independent components, avoiding redundancy and information loss. Through mathematical derivation and geometric visualization, we validate the effectiveness of the approach. (3) Our method significantly improves the baseline by a large margin on four benchmark datasets, achieving state-of-the-art performance on MSR-VTT(52.4%), DiDeMo(53.1%), MSVD(54.0%), and ActivityNet(49.6%).

## 2 RELATED WORKS

**Text-Video Retrieval**    Text-video retrieval is a fundamental task in video understanding. Many studies in this domain can mainly be categorized into three main directions: feature extraction,

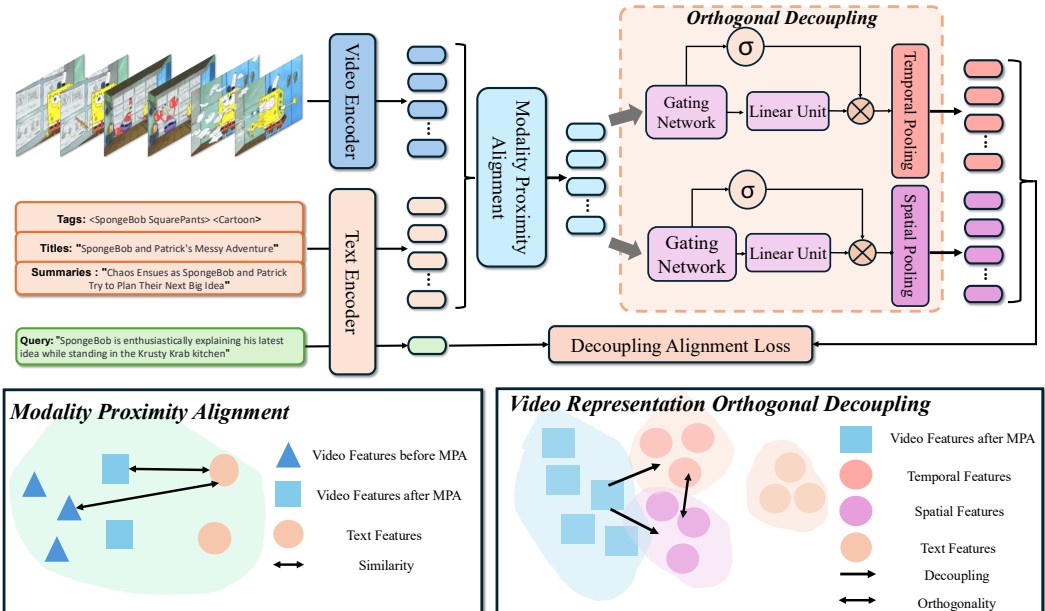

Figure 2: The figure illustrates the architecture of our method, MBDA. The upper flowchart depicts the processing of video features through the MPA and orthogonal decoupling modules. The bottom-left figure shows the effect of the MPA process, where video features are brought closer to text features. The bottom-right figure shows the video features are divided into two orthogonal (non-overlapping) components, which show reduced distribution ranges compared to the original.

feature alignment and matching, objectives functions. For feature extractors, early methods employed separate extractors for the video and text modalities, such as CNN and its variations (Guo et al., 2021; Hu et al., 2018; Qiu et al., 2019; Miech et al., 2018) for video extractors and RNNs (Otani et al., 2019; Wu et al., 2021), Transformers (Gabeur et al., 2020; Patrick et al., 2020; Liu et al., 2021) for text extractors. Since the emergence of contrastive pretrained vision-language encoder, CLIP (Radford et al., 2021), recent methods (Luo et al., 2021; Tang et al., 2025; Gao et al., 2021; Cao et al., 2024; Liu et al., 2023a) primarily focus on fine-tuning CLIP-based models on text-video datasets. Feature alignment and matching aim to project video and text representations into a joint semantic space and compute their similarity. Some former works (Miech et al., 2017; Luo et al., 2020) mainly align on the global features without considering the local details. And recent studies (Wang et al., 2022; Li et al., 2023; Guan et al., 2023) begin to align the feature across the modalities in more fine-grained ways. The existing methods use different objective functions, among which the contrastive loss such as InfoNCE (Oord et al., 2018) has become the most commonly used function. Other loss functions such as triplet loss (Gabeur et al., 2020; Wang et al., 2021; Han et al., 2023) are also used in this task.

**Text-Video Alignment** To address the modality gap between video and text, researchers have explored various strategies for better alignment recent years. CLIP4Clip (Luo et al., 2021) use an aggregate module to get the feature of all frames and then perform the cross-modal alignment. To get more cross-modal correspondences for alignment, many approaches of multi-grained alignment (Zhang et al., 2018; Chen et al., 2020; Yang et al., 2021; Ma et al., 2022) are proposed, which can be divided into mainly two categories: coarse-grained alignment and fine-grained alignment. Coarse-grained alignment (Zhu & Yang, 2020; Li et al., 2021; Dzabraev et al., 2021; Dou et al., 2022) utilizes frame-level or video-level features to match text features, while fine-grained approaches (Wray et al., 2019; Zou et al., 2022; Messina et al., 2021) leverage patch-level features for more precise alignment with text. Some alignment methods combine the above approaches, such as DRL (Wang et al., 2022) using both frame-wise and channel-wise approaches. Existing alignment methods mainly perform a multi-grained split of the input videos and then find the correspondence between the split video and text representations. This makes the alignment vulnerable to the imbalance in modality information capacity and the highly entangled nature of video information. Unlike the aforementioned methods, we propose an alignment approach by pulling video features closer to text and then decoupling them.

## 3 METHOD

In this section, we introduce the novel text-video retrieval approach, Modality-Balanced Decoupling Alignment, as shown in the Figure 2. We first introduce a Modality Proximity Alignment (MPA) module that brings video representations closer to textual representations in the embedding space. To further decouple the large amount of video information, we propose an orthogonal decomposition strategy and demonstrate its orthogonality via both frequency-domain projection and t-SNE (Maaten, 2014) visualization. The decoupled video components retain a smaller proportion of the original information, resulting in a more balanced representation relative to the textual features. Finally, we perform modality-balanced alignment between the decoupled video and textual representations.

### 3.1 PRELIMINARY

**Feature Extraction** For the video modality, a video $v$ is represented as a sequence of sampled frames $v = \{v_1, v_2, \ldots, v_t\}$, where $t$ denotes the number of frames. We employ ViT as the visual encoder backbone. Each frame is divided into non-overlapping patches, which are linearly projected into a 1D token space. A special [CLS] token is prepended to the patch tokens, and all tokens are jointly processed by a Transformer to capture global and local spatial relationships. As a result, the embedding of each frame is represented as a set of tokens $\{z_{\text{cls}}, z_1, \ldots, z_{p^2}\} \in \mathbb{R}^{(1+p^2) \times d}$, where $p \times p$ is the number of spatial patches and $d$ is the token embedding dimension. The sequence of frame embeddings for video $v$ is thus represented as a tensor of shape $t \times (1 + p^2) \times d$. For the text modality, we adopt CLIP's text encoder, which is also based on a Transformer architecture. The activation corresponding to the [EOS] token in the final layer is used as the text embedding. Given a caption $c$, its representation is denoted as $w \in \mathbb{R}^d$.

**Discrete Fourier Transform** The Discrete Fourier Transform (DFT) provides a frequency-domain representation of discrete signals and is widely used for analyzing structural properties such as periodicity, redundancy, and orthogonality. Given a discrete sequence $x = \{x_0, x_1, \ldots, x_{N-1}\} \in \mathbb{R}^N$, its DFT is defined as:

$$\hat{x}_k = \sum_{n=0}^{N-1} x_n \, e^{-i2\pi kn/N}, \quad k = 0, 1, \ldots, N-1, \tag{1}$$

where $i$ denotes the imaginary unit.

Additionally, the Nyquist–Shannon sampling theorem (Shannon, 1949) is an essential principle for digital signal processing, which states that the sample rate must be at least twice the bandwidth of the signal to avoid aliasing. Otherwise, the signal information in the frequency domain will be significantly attenuated due to aliasing.

**Motivation** Video-text retrieval faces a fundamental challenge due to the inherent asymmetry between the two modalities: video features are highly coupled and information-dense, while text features are concise and abstract. Direct alignment in a shared embedding space often leads to redundancy in video representations or loss of fine-grained semantics. To mitigate this, we propose to orthogonally decompose video features into complementary subspaces, ensuring more efficient and semantically balanced alignment with text.

### 3.2 PROPOSED METHOD: MODALITY-BALANCED DECOUPLING ALIGNMENT

In this work, we propose a strategy to address the inherent asymmetry between video and text modalities for retrieval. First, we preprocess the video representations to bring their distribution closer to that of the textual space, which facilitates more effective feature alignment in the subsequent steps. Building on this, we further decouple the video features into orthogonal components, yielding a more balanced semantic space comparable to that of text, thereby enhancing retrieval performance.

**Query-Text Retrieval** In real-world video platforms, it is common practice to associate each video with various forms of textual metadata, such as tags, titles, subtitles, and summaries. This approach appears to be the simplest way to narrow the gap between video and text representations. Moreover, this setup naturally balances the information density and feature place distribution of two modalities by replacing video representations with their textual counterparts. Thus we adopt this method as our baseline, where each video is represented by its associated text. Given a user's query, the retrieval task is formulated as a query-to-text matching problem, where the query is directly compared with the stored textual representation. Formally, each video $V$ is represented by textual data $H(V)$, and the similarity score $\text{Sim}(q, H(V))$ is used to rank candidates.

**Modality Proximity Alignment**    Although the baseline method above successfully pulls the video embedding space to that of the text, it inevitably suffers from information loss: the textual data, as a compressed abstraction of the video content, often omits fine-grained visual details that are crucial for accurately matching diverse user queries. As a result, retrieval performance degrades when queries target specific scenes, objects, or temporal dynamics not explicitly covered in the text data.

To address this limitation, we prove the method above by refining the video representation through incorporating both the global semantics from the textual data $H(V)$ and the rich local details from the original video frames. Specifically, given the video feature sequence $V = \{v_1, v_2, \dots, v_n\}$ and the text feature $H(V)$, we perform a feature-fusion module where video features attend to text data features to incorporate textual semantic information. Specifically, the fused video features are computed by:

$$\tilde{v}_i = \sum_{j=1}^{N} \frac{\exp\left(\langle W_Q^V v_i, W_K^H h_j \rangle / \sqrt{d}\right)}{\sum_{k=1}^{N} \exp\left(\langle W_Q^V v_i, W_K^H h_k \rangle / \sqrt{d}\right)} (W_V^H h_j), \tag{2}$$

where $W_Q^V$, $W_K^H$, and $W_V^H$ are learnable projection matrices, $h$ denotes the components of textual data $H(V)$, $d$ is the hidden dimension, and $\langle \cdot, \cdot \rangle$ denotes the inner product. Finally, a feed-forward network (FFN) and residual connection are applied to obtain the final video representation:

$$V' = \text{LayerNorm}\left(\text{FFN}(\tilde{V}) + V\right), \tag{3}$$

where $\tilde{V} = \{\tilde{v}_1, \tilde{v}_2, \dots, \tilde{v}_n\}$.

This enriched representation better preserves the essential visual information needed for fine-grained alignment with textual queries, while maintaining a distribution closer to the text space for more effective retrieval.

**Video Representation Orthogonal Decoupling**    We propose a novel method to orthogonally decouple video representations into two complementary branches, enabling efficient spatiotemporal modeling without redundancy. Given a video patch-token feature tensor $\mathbf{X} \in \mathbb{R}^{n \times p^2 \times d}$, where $n$ is the number of frames, $p^2$ corresponds to the number of patch tokens, and $d$ is the feature dimension, our goal is to decouple $\mathbf{X}$ into two orthogonal subspaces, each focusing on different spatiotemporal properties. It should be noted that the decoupled patch tokens are individually concatenated with the original [CLS] token, serving as the final visual embeddings for video-text retrieval.

We design two distinct feature branches as follows. We introduce projection functions based on learnable gated linear units (GLUs) to enable more expressive and adaptive feature decomposition. Specifically, we construct two gated branches that project $\mathbf{X}$ into temporally and spatially enhanced representations using independent linear projections followed by GLUs. The temporal-enhanced branch maintains the original frame rate $n$ while reducing the spatial resolution by a factor of $\lambda$ across patch tokens. The spatial-enhanced branch maintains full spatial resolution while reducing the temporal resolution by a factor of $\mu$.

$$\mathbf{X}_{\text{temporal}} = \text{GLU}_{\text{tem}}(\mathbf{X}) \in \mathbb{R}^{n \times (p/\lambda)^2 \times d}. \tag{4}$$

$$\mathbf{X}_{\text{spatial}} = \text{GLU}_{\text{spa}}(\mathbf{X}) \in \mathbb{R}^{(n/\mu) \times p^2 \times d}. \tag{5}$$

Our MBDA method is primarily presented above through the use of MPA and VROD modules. In the following theoretical analysis, we further demonstrate the orthogonality of the decoupled branches.

**Theoretical Analysis via Discrete Fourier Transform**    To derive the orthogonality between the two branches, we model $\mathbf{X}$ under a spatiotemporally separable low-rank decomposition:

$$\mathbf{X}(t, x) = \mathbf{T}(t) \cdot \mathbf{S}(x)^\top, \quad x \in [0, p^2], \tag{6}$$

where $\mathbf{T}(t) \in \mathbb{R}^{n \times \delta \times d}$ captures temporal structures, and $\mathbf{S}(x) \in \mathbb{R}^{p^2 \times \delta \times d}$ captures spatial structures. $\delta$ represents the rank of matrix $\mathbf{X}$. We separately apply the decomposition (6) on both branches($\mathbf{X}_{\text{spatial}}$ and $\mathbf{X}_{\text{temporal}}$), getting four decomposed matrix: $T_{temporal}$, $S_{temporal}$, $T_{spatial}$, and $S_{spatial}$.

Table 1: Text-to-video retrieval performance on MSR-VTT, DiDeMo, MSVD, and ActivityNet. Bold denotes the best performance. "–" denotes that the result is unavailable.

(a) MSR-VTT Retrieval

| Method | R@1 ↑ | R@5 ↑ | R@10 ↑ | MdR ↓ | MnR ↓ |
|---|---|---|---|---|---|
| CLIP-ViT-B/32 | | | | | |
| ProST (Li et al., 2023) | 48.2 | 74.6 | 83.4 | 2.0 | 12.4 |
| DiffusionRet (Jin et al., 2023) | 49.0 | 75.2 | 82.7 | 2.0 | 12.1 |
| UCOFIA (Wang et al., 2023) | 49.4 | 72.1 | 83.5 | 2.0 | 12.9 |
| TEFAL (Ibrahimi et al., 2023) | 49.4 | 75.9 | 83.9 | 2.0 | **12.0** |
| Cap4Video (Wu et al., 2023) | 49.3 | 74.3 | 83.8 | 2.0 | **12.0** |
| TeachCLIP (Tian et al., 2023) | 45.6 | 71.9 | 81.8 | – | – |
| AVIGATE (Jeong et al., 2025) | 50.2 | 74.3 | 83.2 | – | – |
| TempMe (Shen et al., 2024) | 46.1 | 71.8 | 80.7 | – | 14.8 |
| MBDA (Ours) | **52.4** | **75.9** | **85.4** | **1.0** | 13.2 |
| CLIP-ViT-B/16 | | | | | |
| X-CLIP (Ma et al., 2022) | 49.3 | 75.8 | 84.8 | 2.0 | 12.2 |
| ProST (Li et al., 2023) | 49.5 | 75.0 | 84.0 | 2.0 | 11.7 |
| UCOFIA (Wang et al., 2023) | 49.8 | 74.6 | 83.5 | 2.0 | 13.3 |
| Cap4Video (Wu et al., 2023) | 51.4 | 75.7 | 83.9 | 1.0 | 12.4 |
| AVIGATE (Jeong et al., 2025) | 52.1 | 76.4 | 85.2 | – | – |
| TempMe (Shen et al., 2024) | 49.0 | 74.4 | 83.3 | – | – |
| MBDA (Ours) | **52.6** | **76.5** | **84.4** | **1.0** | **11.6** |

(b) ActivityNet Retrieval

| Method | R@1 ↑ | R@5 ↑ | R@10 ↑ | MdR ↓ | MnR ↓ |
|---|---|---|---|---|---|
| CLIP-ViT-B/32 | | | | | |
| DRL (Wang et al., 2022) | 44.2 | 74.5 | 86.1 | 2.0 | – |
| DiffusionRet (Jin et al., 2023) | 45.8 | 75.6 | 86.3 | 2.0 | 6.5 |
| X-CLIP (Ma et al., 2022) | 44.3 | 74.1 | – | – | 7.9 |
| UCOFIA (Wang et al., 2023) | 45.7 | 76.6 | 86.6 | 2.0 | 6.4 |
| CenterCLIP (Zhao et al., 2022) | 43.9 | 75.3 | 85.2 | 2.0 | 7.0 |
| TempMe (Shen et al., 2024) | 44.9 | 75.2 | 85.5 | – | 6.8 |
| MBDA (Ours) | **49.6** | **79.4** | **89.3** | **2.0** | **5.2** |
| CLIP-ViT-B/16 | | | | | |
| DRL (Wang et al., 2022) | 46.2 | 77.3 | 88.2 | 2.0 | – |
| X-CLIP (Ma et al., 2022) | 46.2 | 75.5 | – | – | 6.8 |
| CenterCLIP (Zhao et al., 2022) | 46.2 | 77.0 | 87.6 | 2.0 | 5.7 |
| MBDA (Ours) | **51.8** | **81.2** | **90.5** | **1.0** | **4.5** |

(c) DiDeMo Retrieval

| Method | R@1 ↑ | R@5 ↑ | R@10 ↑ | MdR ↓ | MnR ↓ |
|---|---|---|---|---|---|
| CLIP-ViT-B/32 | | | | | |
| X-CLIP (Ma et al., 2022) | 45.2 | 74.0 | – | – | 14.6 |
| DRL (Wang et al., 2022) | 47.9 | 73.8 | 82.7 | 2.0 | – |
| STAN (Liu et al., 2023b) | 46.5 | 71.5 | 80.9 | 2.0 | – |
| ProST (Li et al., 2023) | 44.9 | 72.7 | 82.7 | 2.0 | 13.7 |
| DiffusionRet (Jin et al., 2023) | 46.7 | 74.7 | 82.7 | 2.0 | 14.3 |
| UCOFIA (Wang et al., 2023) | 46.5 | 74.8 | 84.4 | 2.0 | 13.4 |
| Cap4Video (Wu et al., 2023) | 52.0 | **79.4** | **87.5** | 1.0 | 10.5 |
| TempMe (Shen et al., 2024) | 48.0 | 72.4 | 81.8 | – | – |
| MBDA (Ours) | **53.1** | 78.7 | 86.5 | **1.0** | **10.4** |
| CLIP-ViT-B/16 | | | | | |
| X-CLIP (Ma et al., 2022) | 47.8 | 79.3 | – | – | 12.6 |
| DRL (Wang et al., 2022) | 49.0 | 76.5 | 84.5 | 2.0 | – |
| ProST (Li et al., 2023) | 47.5 | 75.5 | 84.4 | 2.0 | 12.3 |
| MBDA (Ours) | **54.3** | **79.3** | **87.6** | **1.0** | **9.8** |

(d) MSVD Retrieval

| Method | R@1 ↑ | R@5 ↑ | R@10 ↑ | MdR ↓ | MnR ↓ |
|---|---|---|---|---|---|
| CLIP-ViT-B/32 | | | | | |
| DRL (Wang et al., 2022) | 48.3 | 79.1 | 87.3 | 2.0 | – |
| X-Pool (Gorti et al., 2022) | 47.2 | 77.4 | 86.0 | 2.0 | 9.3 |
| DiffusionRet (Jin et al., 2023) | 46.6 | 75.9 | 84.1 | 2.0 | 15.7 |
| X-CLIP (Ma et al., 2022) | 47.1 | 77.8 | – | – | 9.5 |
| UCOFIA (Wang et al., 2023) | 47.4 | 77.6 | – | – | 9.6 |
| CenterCLIP (Zhao et al., 2022) | 47.6 | 77.5 | 86.0 | 2.0 | 9.8 |
| Cap4Video (Wu et al., 2023) | 51.8 | 80.8 | **88.3** | 1.0 | **8.3** |
| TeachCLIP (Tian et al., 2023) | 47.4 | 77.3 | – | – | – |
| MBDA (Ours) | **54.0** | **81.6** | 88.2 | **1.0** | 8.7 |
| CLIP-ViT-B/16 | | | | | |
| UATVR (Fang et al., 2023) | 49.7 | 79.0 | 87.3 | 2.0 | 8.9 |
| DRL (Wang et al., 2022) | 50.0 | 81.5 | **89.5** | 2.0 | – |
| X-CLIP (Ma et al., 2022) | 50.4 | 80.6 | – | – | 8.4 |
| CenterCLIP (Zhao et al., 2022) | 50.6 | 80.3 | 88.4 | 1.0 | 8.4 |
| MBDA (Ours) | **55.1** | **82.8** | 89.2 | **1.0** | **8.1** |

To analyze their orthogonality, we project the components into the frequency domain via the Discrete Fourier Transform (DFT), separately along the temporal and spatial axes as follows:

$$\mathcal{F}_t(\mathbf{X}_{\mathbf{t.}/\mathbf{s.}})(\omega, x) = \sum_{t=0}^{n-1} \mathbf{X}_{\mathbf{t.}/\mathbf{s.}}(t, x) e^{-i2\pi\omega t/n} \quad (7)$$

$$\mathcal{F}_x(\mathbf{X}_{\mathbf{t.}/\mathbf{s.}})(t, \xi) = \sum_{x=0}^{p^2-1} \mathbf{X}_{\mathbf{t.}/\mathbf{s.}}(t, x) e^{-i2\pi\xi x/p^2}, \quad (8)$$

and utilizing the separability assumption, we apply a 2D joint discrete Fourier transform (DFT) along the temporal axis $t \in [0, n-1]$ and spatial axis $x \in [0, p^2 - 1]$:

$$\mathcal{F}_{t,x}(\mathbf{X}_{\mathbf{t.}/\mathbf{s.}})(\omega, \xi) = \sum_{t=0}^{n-1}\sum_{x=0}^{p^2-1} \mathbf{X}_{\mathbf{t.}/\mathbf{s.}}(t, x) \cdot e^{-i2\pi\omega t/n} \cdot e^{-i2\pi\xi x/p^2} \xrightleftharpoons[\text{Serialization}]{\text{Matrixization}} \mathcal{F}_t(\mathbf{T}_{\mathbf{t.}/\mathbf{s.}})(\omega) \cdot \mathcal{F}_x(\mathbf{S}_{\mathbf{t.}/\mathbf{s.}})(\xi)^\top \quad (9)$$

Specifically, the temporal branch retains the original temporal resolution while performing spatial downsampling, and vice versa for the spatial branch. In the frequency domain, downsampling introduces aliasing effects, high-frequency components are lost. When the sampling rate is sufficiently low, most of the information in the frequency domain can be considered lost. This is exactly what $T_{\text{spatial}}$ and $S_{\text{temporal}}$ undergo, while $T_{\text{temporal}}$ and $S_{\text{spatial}}$ retain the full frequency-domain information:

$$\langle \mathcal{F}(T_{\text{temporal}}), \mathcal{F}(T_{\text{spatial}})\rangle = 0, \quad \langle \mathcal{F}(S_{\text{temporal}}), \mathcal{F}(S_{\text{spatial}})\rangle = 0$$

Table 2: Ablation study of the Modality Proximity Adjustment (MPA) and Video Representation Orthogonal Decoupling (VROD) modules on the MSR-VTT dataset.

| Method | R@1 ↑ | R@5 ↑ | R@10 ↑ | MdR ↓ | MnR ↓ |
|---|---|---|---|---|---|
| Video features for retrieval | 45.8 | 72.1 | 79.6 | 2.0 | 15.5 |
| Query-Text Retrieval (QTR) Baseline | 42.1 | 67.3 | 75.9 | 2.0 | 20.5 |
| MPA only | 49.2 | 74.5 | 83.2 | 2.0 | 13.6 |
| VROD only | 47.8 | 73.5 | 80.7 | 2.0 | 14.6 |
| **MPA + VROD (Full Model)** | **52.4** | **75.9** | **85.4** | **1.0** | **13.2** |

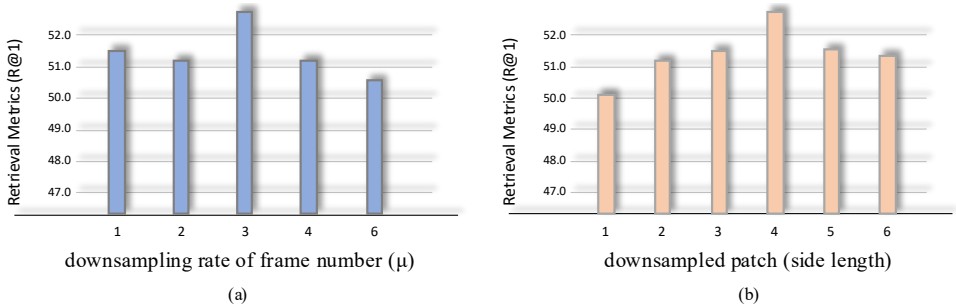

(a)                          (b)

Figure 3: Figure (a) and (b) shows the comparison of frame downsampling rate ($\mu$) and downsampled patch token size on retrieval performance (R@1). Best performance is achieved at $\mu = 3$ and a patch size of $4 \times 4$.

$$\xrightarrow{(9)} \langle \mathcal{F}_{t,x}(\mathbf{X}_{\text{temporal}}), \mathcal{F}_{t,x}(\mathbf{X}_{\text{spatial}}) \rangle = 0 \tag{10}$$

Based on the above derivation, we can conclude that $\mathbf{X}_{\text{temporal}}$ and $\mathbf{X}_{\text{spatial}}$ are orthogonal in the frequency domain, ensuring that our decomposition does not introduce redundant information. We also employ t-SNE (Maaten, 2014) visualizations in Section 4.3 and quantitative calculation of cosine similarities in Appendix A.5 to illustrate the orthogonality.

**Orthogonal Decoupling Alignment Loss**     We employ a unified cross-entropy retrieval loss

$$\mathcal{L}_{v \to t} = -\frac{1}{N} \sum_{i=1}^{N} \log \frac{\exp(s(t_i, v_i)/\lambda)}{\sum_{j=1}^{N} \exp(s(t_j, v_i)/\lambda)} \tag{11}$$

$$\mathcal{L}_{t \to v} = -\frac{1}{N} \sum_{i=1}^{N} \log \frac{\exp(s(t_i, v_i)/\lambda)}{\sum_{j=1}^{N} \exp(s(t_i, v_j)/\lambda)} \tag{12}$$

$$\mathcal{L}_{\text{CE}}(t, v) = \frac{1}{2}(\mathcal{L}_{t \to v} + \mathcal{L}_{v \to t}). \tag{13}$$

Let $V_{\text{temporal}}$ and $V_{\text{spatial}}$ be the orthogonally decoupled video branches added with [CLS] tokens respectively. We define their combined alignment loss as

$$\mathcal{L}_{\text{align}} = \alpha \, \mathcal{L}_{\text{CE}}(q, V_{\text{temporal}}) + \beta \, \mathcal{L}_{\text{CE}}(q, V_{\text{spatial}}), \tag{14}$$

with weights $\alpha, \beta > 0$. To stay consistent with the query-text retrieval (Sec. **3.2**), we also include

$$\mathcal{L}_{\text{T}} = \mathcal{L}_{\text{CE}}\big(q, H(V)\big). \tag{15}$$

The final training objective is

$$\mathcal{L}_{\text{total}} = \mathcal{L}_{\text{align}} + \mathcal{L}_{\text{T}}, \tag{16}$$

## 4   EXPERIMENTS

We conduct our experiments on four popular text-video retrieval datasets, i.e., MSR-VTT (Xu et al., 2016), MSVD (Wu et al., 2016), DiDeMo (Hendricks et al., 2017), and ActivityNet (Caba Heilbron et al., 2015). The detailed experiment settings are shown in the appendix A.1. Additionally, we show the experiments on video-to-text retrieval, experiments with post-processing techniques, additional ablation studies, qualitative and quantitative results in the appendix.

Table 3: Performance comparison with different weight ($\alpha$ and $\beta$) allocations between the two decoupled branches.

| $\alpha$ | $\beta$ | R@1 $\uparrow$ |
|---|---|---|
| 1.0 | 0.0 | 50.7 |
| 0.7 | 0.3 | 51.2 |
| 0.5 | 0.5 | **52.4** |
| 0.3 | 0.7 | 50.6 |
| 0.0 | 1.0 | 49.7 |

Table 4: Ablation study on the effectiveness of each loss component. The combination of both losses achieves the best performance.

| $\mathcal{L}_{\text{align}}$ | $\mathcal{L}_{\text{T}}$ | R@1 $\uparrow$ |
|---|---|---|
| $\checkmark$ | | 51.1 |
| | $\checkmark$ | 42.1 |
| $\checkmark$ | $\checkmark$ | **52.4** |

### 4.1 COMPARISONS WITH STATE-OF-THE-ART

We compare the text-to-video retrieval performance of MBDA with recent state-of-the-art methods on four benchmark datasets. Comparisons on MSR-VTT, DiDeMo, MSVD and ActivityNet are shown in Table 1. The proposed MBDA can achieve the best performance on different datasets. For example, on MSR-VTT, our method outperforms the AVIGATE by 2.2% at R@1 on CLIP-ViT-B/32. On DiDeMo, MBDA outperforms Cap4Video by 1.1% with CLIP-ViT-B/32. On the MSVD dataset, where each video is associated with multiple captions under different evaluation process, our method also achieves the best performance, outperforming Cap4Video by 2.2%. On larger-scale video datasets such as ActivityNet, our method still maintains a significant performance lead, outperforming DiffusionRet by 3.8%. Moreover, our method also performs well on larger CLIP backbones such as CLIP-ViT-B/16, outperforming previous SOTA methods by 0.5%, 5.3%, 4.5%, and 5.6% on MSR-VTT, DiDeMo, MSVD, and ActivityNet respectively.

### 4.2 ABLATION STUDY

**Model Components** We provide an ablation study on MSR-VTT in terms of Modality Proximity Adjustment module and Video Representation Orthogonal Decoupling in Table 2. First, we show the baseline method Query-Text Retrieval, where we use text summaries to take place the video representations for retrieval. To improve this baseline method, we use a MPA module to preserve the video information, while at the same time maintaining the privilege of QTR, pulling video representation closer to that of the query. By comparison, we obtain 7.1% boost at R@1. Also, this also outperforms the method where the video representations before MPA processing are used for retrieval. We further add the decoupling module, and get 3.2% boost at R@1 comparing to the experiment with MPA. We also find that without MPA module, the decoupling module cannot fully demonstrate its effectiveness solely. This is because, without first pulling video features closer to the text feature space, the decoupled video components cannot be clearly separated in the frequency domain where text representations serve as the basis. As a result, both branches processed by only VROD module fail to align well with the textual queries, leading to suboptimal improvement.

**Analysis for Decouple Module** *Effect of downsampling rates* The parameter $\mu$, which means the downsampling rate of frame number, we evaluate the scale range setting $\mu \in [1, 6]$ as shown in Figure 3(a). Also we evaluate the downsampled patch token amount from $1 \times 1$ to $6 \times 6$, as shown in Figure 3(b). We find that R@1 preforms best when $\mu = 3$ and the downsampled patch token amount is $4 \times 4$. Subsequent experiments confirm that this setting of downsampling rate is consistently optimal across all four datasets tested.

*Effect of the weights of two branches* We further investigate the impact of the weight allocation between the two decoupled branches on the MSR-VTT dataset. As shown in Table 3, assigning equal weights (0.5 for each branch) achieves the best overall performance, indicating that balancing the contributions of both branches is crucial for optimal results. Subsequent experiments confirm that this weight setting is consistently optimal across all four datasets tested.

**Analysis for Loss Function** We conduct ablation studies on the effectiveness of each component in the total loss function, defined as $\mathcal{L}_{\text{total}} = \mathcal{L}_{\text{align}} + \mathcal{L}_{\text{T}}$. Specifically, we compare the performance when using only $\mathcal{L}_{\text{align}}$, only $\mathcal{L}_{\text{T}}$, and the combination of both on the MSR-VTT dataset. As shown in Table 4, incorporating both loss components leads to consistently better results, indicating that each part contributes positively to the overall performance.

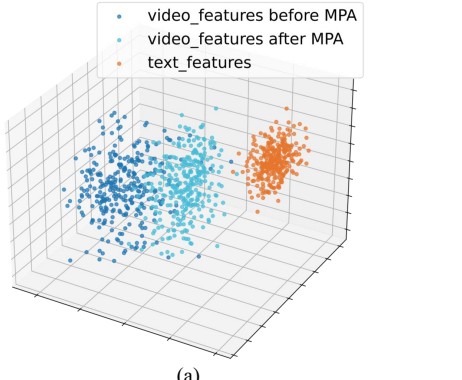 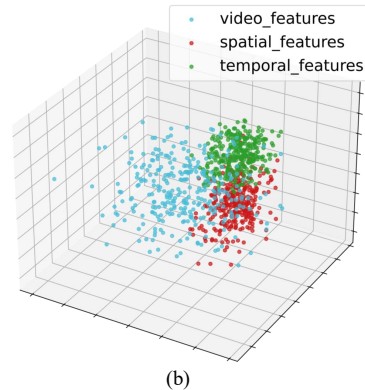

(a)                      (b)

Figure 4: This figure visualizes the impact of the Modality Proximity Alignment (MPA) module and the Video Representation Orthogonal Decoupling (VROD) module on the distribution of video and text features using t-SNE (Maaten, 2014). (a) shows how the MPA aligns video features closer to text features in vector space, enhancing their proximity. (b) shows the VROD's ability to decouple video features into orthogonal spatial and temporal components, which results in a more balanced distribution relative to the original video features, thereby improving alignment with text features.

### 4.3 DOES MBDA EFFECTIVELY DECOUPLE THE VIDEO REPRESENTATION?

We further analyze the behavior of our MBDA method by examining the distribution of text and video representations in the embedding space through visualization and computing the similarity between decoupled components. As shown in Figure 4(a), we employ t-SNE (Maaten, 2014) to visualize the embeddings of the text representations $t_i$, as well as the video representations before and after the MPA module, i.e., $v_i$ and $v_i'$. We observe that the distance between the two modalities is effectively reduced after applying the MPA module. Furthermore, we visualize the transformed video embeddings $v_i'$ and the two branches obtained after the decoupling module, denoted as $(v_t')_i$ and $(v_s')_i$. As illustrated in Figure 4(b), the original video embedding $v_i'$, which initially spans a relatively large space, is decomposed into two components that occupy significantly smaller and more compact subspaces. To quantitatively verify this result, we compute the orthogonality of these resulting components in our experiments in Appendix A.5, where we show their average cosine similarity approaches zero. This decomposition mitigates the modality gap caused by the inherent information imbalance between video and text. In addition, the minimal overlap between the two disentangled components further supports the orthogonality and non-redundancy of our decomposition strategy.

Furthermore, to demonstrate their semantic meaningfulness, we conduct a case study in Appendix A.5 showing that the temporal and spatial components align better with corresponding motion- and scene-related text queries, respectively. Finally, to confirm that our method enhances alignment without information loss, we also show in Appendix A.5 that the average similarity between the video components and text embeddings is significantly higher than that of the baseline.

## 5 CONCLUSION

In this work, we proposed MBDA, a novel text-video retrieval method by decoupling the video embedding into orthogonal components, to address the challenge of multi-modal alignment in text-video retrieval. The primary contribution of this work lies in introducing a strategy to orthogonally decouple the complex semantics in video representations, thereby addressing the imbalance between the feature spaces of video and text. We identified two key challenges in text-video retrieval: (1) an imbalance in the semantic capacity of video and text representations—videos often contain rich information not described by text, making alignment difficult; (2) the highly entangled nature of information within video representations, which hinders the direct mapping or correspondence with text representations. To tackle these issues, MBDA brings video and text representations closer in feature space to facilitate disentanglement and alignment, and further decouples the video representation into two orthogonal components. These orthogonal representations are more semantically balanced with respect to text features, thereby enabling more effective alignment. Extensive experiments demonstrate that MBDA achieves state-of-the-art performance across multiple benchmark datasets. We hope MBDA can inspire future research on modality balance in the text-video retrieval.

## 6 ETHICS STATEMENT

**Data Usage**  Our experiments were conducted exclusively on publicly available, well-established benchmark datasets: MSR-VTT, DiDeMo, MSVD, and ActivityNet. These datasets are widely used by the research community for academic purposes and were collected and distributed by their original creators in accordance with their respective data usage policies. We did not collect any new data for this study, nor did we involve human subjects in our experiments.

**Potential Societal Impacts**  We acknowledge that advancements in retrieval technology, including our own, could potentially be misused for applications such as the retrieval of sensitive or private information if applied to non-public data sources. However, the primary goal of our research is to advance cross-modal understanding, which has numerous beneficial applications, including improved accessibility for visually impaired users, enhanced educational tools, and more efficient content discovery on public platforms. We believe that the positive applications of this technology outweigh the potential risks, and we encourage the community to continue developing responsible AI practices and safeguards alongside technological progress. Our work does not introduce any inherent biases beyond those that may already exist in the benchmark datasets, and we have not designed it for any malicious purposes.

## 7 REPRODUCIBILITY STATEMENT

We are committed to ensuring the reproducibility of our research. To this end, we have made the following efforts.

**Code**  The source code for our model, is zipped in the Supplementary Material. This will allow for the complete replication of our experiments and results.

**Implementation Details**  We provide comprehensive implementation details in both main text and Appendix. Thess sections cover the specific settings for all datasets, including batch size, learning rates, optimizer parameters, and the number of training epochs.

We believe these resources provide a clear and sufficient pathway for the research community to reproduce our findings and build upon our work.

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

# A APPENDIX

We provide in-depth discussions, more results of the proposed MBDA as follows:

- Some implementation details of our experiments.
- Experiments on video-to-text retrieval on four datasets.
- Experiments with post-processing techniques.
- Some quantitative results.
- Some qualitative results.
- About our use of LLMs.

## A.1 SOME IMPLEMENTATION DETAILS

**Datasets and Evaluation Metris** We conduct our experiments on four popular text-video retrieval datasets, i.e., MSR-VTT (Xu et al., 2016), MSVD (Wu et al., 2016), DiDeMo (Hendricks et al., 2017), and ActivityNet (Caba Heilbron et al., 2015). Following common practices (Luo et al., 2021; Wang et al., 2022; Jin et al., 2022), we preprocess these datasets to ensure fair comparisons across experiments. We adopt standard retrieval metrics, including Recall at rank K (R@K), median rank (MdR) and mean rank (MnR) as metrics, to evaluate the performance of text-video and video-text retrieval tasks. MSR-VTT (Xu et al., 2016) is the most popular benchmark consisting of 10,000 YouTube video clips from 20 categories, and each video clip is annotated with 20 English sentences. We follow the 1k-A split (Liu et al., 2019) with 9,000 videos for training and 1,000 for testing. MSVD (Wu et al., 2016) includes 1,970 videos with 80,000 captions. We report results on split set, where train, validation and test are 1200, 100 and 670 videos. DiDeMo (Hendricks et al., 2017) contains 10,000 videos annotated with 40,000 sentences. We use the training/testing data following (Jin et al., 2023). ActivityNet (Caba Heilbron et al., 2015) consists of 20,000 YouTube videos. We concatenate all descriptions of a video to a single query and evaluate on the 'val1' split (10,009 training, 4,917 testing).

**Implementation Settings** We utilize the CLIP (ViT-B/32) (Radford et al., 2021) equipped with Temporal Transformer (Luo et al., 2021) as pre-trained Bi-Encoder. Following the setup in previous work (Luo et al., 2021), we finetune our model using frame number of 12 and word number of 32 for MSR-VTT and MSVD, and 64, 64 for DiDeMo and ActivityNet. The batch size is set to 256 for MSR-VTT and MSVD, 64 for ActivityNet and DiDeMo. We adopt Adam optimizer (Kingma & Ba, 2014) with a linear warmup as our optimizer, with warm-up proportion of 0.1, and weight decay of 0.2. The learning rate is set as 1e-7 for CLIP model and the gate linear unit modules, and 1e-4 for non-CLIP modules. The training epochs is set to 5 for all datasets. During our decoupling step, we set the downsample rate $\mu$ for the number of frames is 3. And we downsample the patch token from $7 \times 7$ to $4 \times 4$. The weight of both branches $(\alpha, \beta)$ is 0.5 and 0.5. The weight of $\mathcal{L}_{\mathrm{align}}$ and $\mathcal{L}_{\mathrm{T}}$ $(\theta, \gamma)$ is 0.6, 0.4 for MSR-VTT, 0.5, 0.5 for MSVD and 0.7, 0.3 for other datasets. We implement our experiments with PyTorch on 8 NVIDIA V100 GPUs.

For datasets like MSR-VTT, where each video in the training set is associated with multiple captions, we add a portion of the captions from the training set as textual data in the Modality Proximity Alignment module. This approach achieves competitive performance. We also adopt the LLaVA-Video-7B model (Zhang et al., 2024) to generate textual data on the four datasets. Specifically, we input only the video frames from the datasets into the model, without any query data, to obtain summaries of the videos.

**Training Resources Usage** Our experiments are performed on 8 NVIDIA V100 GPUs. The batch size is set to 256 for MSR-VTT and MSVD, 64 for ActivityNet and DiDeMo on CLIP-ViT-B/32. We should decrease the batch size to a quarter when we perform on CLIP-ViT-B/16. The test dataset of ActivityNet is much larger than that of MSR-VTT. To avoid the CUDA out of memory error, we move the processed video and text features to CPU and place each chunked block back onto the GPU to calculate the similarity blocks. Finally, we concatenated these chunked similarity matrices and finish the evaluation. This processing method saves CUDA memory at the cost of increased training time. The training resources used on different datasets is shown in Table 5.

Table 5: Training Resources Usage

| Dataset | Training Time | GPU Memory Usage (per GPU) |
|---------|---------------|----------------------------|
| MSR-VTT | 4.72 h | 26.63 GiB |
| DiDeMo | 2.23 h | 24.90 GiB |
| MSVD | 3.53 h | 20.73 GiB |
| ActivityNet | 10.59 h | 19.08 GiB |

## A.2 EXPERIMENTS ON VIDEO-TO-TEXT RETRIEVAL

**Experiment Settings**  We conduct our video-to-text retrieval experiments with the same dataset settings as the text-to-retrieval ones. It should be noted that all other experimental settings are the same as those of the text-to-video retrieval experiment.

**Results**  The results of video-to-text experiments on four dataset benchmarks, MSR-VTT, DiDeMo, MSVD and ActivityNet are shown in the Table 6, 7, 8, 9. As the table shows, our MBDA method successfully outperforms other state-of-art approaches on CLIP-ViT-B/32 and CLIP-ViT-B/16.

Table 6: Video-to-text retrieval performance on MSR-VTT. Bold denotes the best performance. "–" denotes that the result is unavailable.

| Method | R@1 ↑ | R@5 ↑ | R@10 ↑ | MdR ↓ | MnR ↓ |
|--------|-------|-------|--------|-------|-------|
| X-CLIP (Ma et al., 2022) | 46.8 | 73.3 | 84.0 | 2.0 | 9.1 |
| X-Pool (Gorti et al., 2022) | 44.4 | 73.3 | 84.0 | 2.0 | 9.0 |
| ProST (Li et al., 2023) | 46.3 | 74.2 | 83.2 | 2.0 | 8.7 |
| DiffusionRet (Jin et al., 2023) | 47.7 | 73.8 | 84.5 | 2.0 | 8.8 |
| UATVR (Fang et al., 2023) | 46.9 | 73.8 | 83.8 | 2.0 | 8.6 |
| UCOFIA (Wang et al., 2023) | 47.1 | 74.3 | 83.0 | 2.0 | 11.4 |
| TEFAL (Ibrahimi et al., 2023) | 47.1 | 75.1 | 84.9 | 2.0 | **7.4** |
| AVIGATE (Jeong et al., 2025) | 49.7 | 75.3 | 83.7 | – | – |
| TempMe (Shen et al., 2024) | 45.6 | 72.4 | 81.2 | – | 10.2 |
| MBDA (Ours) | **51.4** | **77.3** | **85.6** | **1.0** | 8.9 |
| CLIP-ViT-B/16 | | | | | |
| X-CLIP (Ma et al., 2022) | 48.9 | 76.8 | 84.5 | 2.0 | 8.1 |
| ProST (Li et al., 2023) | 48.0 | 75.9 | 85.2 | 2.0 | 8.3 |
| UATVR (Fang et al., 2023) | 48.1 | 76.3 | 85.4 | 2.0 | **8.0** |
| UCOFIA (Wang et al., 2023) | 49.1 | 77.0 | 83.8 | 2.0 | 11.2 |
| AVIGATE (Jeong et al., 2025) | 51.2 | **77.9** | 86.2 | – | – |
| TempMe (Shen et al., 2024) | 47.6 | 75.3 | 85.4 | – | 9.0 |
| MBDA (Ours) | **52.8** | 77.7 | **86.5** | **1.0** | 8.1 |

## A.3 EXPERIMENT WITH POST-PROCESSING TECHNIQUES

We also employ the commonly used post-processing techniques (Cheng et al., 2021; Bogolin et al., 2022) in text-video retrieval, on our method, showing better performance. The experiment result is shown in Table 10.

## A.4 QUALITATIVE RESULTS

We showcase text-to-video retrieval outcomes of the intermediate-structure based method EMCL (Jin et al., 2022) and our MBDA. As shown in Figure 5, our method accurately captures the motion information presented in the image. In Figure 6, it also demonstrates a strong ability to recognize fine-grained visual details.

Table 7: Video-to-text retrieval performance on DiDeMo. Bold denotes the best performance. "–" denotes that the result is unavailable.

| Method | R@1 ↑ | R@5 ↑ | R@10 ↑ | MdR ↓ | MnR ↓ |
|---|---|---|---|---|---|
| CLIP-ViT-B/32 | | | | | |
| X-CLIP (Ma et al., 2022) | 43.1 | 72.2 | – | – | 10.9 |
| DiffusionRet (Jin et al., 2023) | 46.2 | 74.3 | 82.2 | 2.0 | 10.7 |
| UCOFIA (Wang et al., 2023) | 46.0 | 71.9 | 81.5 | 2.0 | 12.1 |
| TempMe (Shen et al., 2024) | 48.4 | 75.4 | 83.6 | – | 9.1 |
| MBDA (Ours) | **51.7** | **79.3** | **86.6** | **1.0** | **7.7** |
| CLIP-ViT-B/16 | | | | | |
| CLIP4Clip (Luo et al., 2021) | 47.2 | 74.0 | – | – | 10.5 |
| DRL (Wang et al., 2022) | 49.9 | 75.4 | 83.3 | 2.0 | – |
| X-CLIP (Ma et al., 2022) | 47.8 | 76.8 | – | – | 10.5 |
| MBDA (Ours) | **52.1** | **80.2** | **88.7** | **1.0** | **7.2** |

Table 8: Video-to-text retrieval performance on MSVD. Bold denotes the best performance. "–" denotes that the result is unavailable.

| Method | R@1 ↑ | R@5 ↑ | R@10 ↑ | MdR ↓ | MnR ↓ |
|---|---|---|---|---|---|
| CLIP-ViT-B/32 | | | | | |
| DRL (Wang et al., 2022) | 62.3 | 86.3 | 92.2 | 1.0 | – |
| X-Pool (Gorti et al., 2022) | 66.4 | 90.0 | 94.2 | 1.0 | 3.3 |
| DiffusionRet (Jin et al., 2023) | 61.9 | 88.3 | 92.9 | 1.0 | 4.5 |
| X-CLIP (Ma et al., 2022) | 60.9 | 87.8 | – | – | 4.7 |
| CenterCLIP (Zhao et al., 2022) | 63.5 | 86.4 | 92.6 | 1.0 | 3.8 |
| MBDA (Ours) | **70.2** | **92.5** | **95.9** | **1.0** | **2.5** |
| CLIP-ViT-B/16 | | | | | |
| DRL (Wang et al., 2022) | 68.7 | 92.5 | 95.6 | 1.0 | – |
| X-CLIP (Ma et al., 2022) | 66.8 | 90.4 | – | – | 4.2 |
| CenterCLIP (Zhao et al., 2022) | 68.4 | 90.1 | 95.0 | 1.0 | 3.0 |
| MBDA (Ours) | **74.3** | **94.0** | **97.1** | **1.0** | **2.1** |

Table 9: Video-to-text retrieval performance on ActivityNet. Bold denotes the best performance. "–" denotes that the result is unavailable.

| Method | R@1 ↑ | R@5 ↑ | R@10 ↑ | MdR ↓ | MnR ↓ |
|---|---|---|---|---|---|
| CLIP-ViT-B/32 | | | | | |
| DRL (Wang et al., 2022) | 42.2 | 74.0 | 86.2 | 2.0 | – |
| DiffusionRet (Jin et al., 2023) | 43.8 | 75.3 | 86.7 | 2.0 | 6.3 |
| X-CLIP (Ma et al., 2022) | 43.9 | 73.9 | – | – | 7.6 |
| UCOFIA (Wang et al., 2023) | 46.3 | 76.5 | 86.3 | 2.0 | 6.7 |
| CenterCLIP (Zhao et al., 2022) | 44.5 | 75.7 | 86.2 | 2.0 | 6.5 |
| TempMe (Shen et al., 2024) | 45.3 | 74.7 | 86.2 | – | 6.4 |
| MBDA (Ours) | **47.4** | **78.8** | **89.3** | **2.0** | **5.1** |
| CLIP-ViT-B/16 | | | | | |
| DRL (Wang et al., 2022) | 45.7 | 76.5 | 87.8 | 2.0 | – |
| X-CLIP (Ma et al., 2022) | 46.4 | 75.9 | – | – | 6.4 |
| CenterCLIP (Zhao et al., 2022) | 46.7 | 77.1 | 88.0 | 2.0 | 5.5 |
| MBDA (Ours) | **49.1** | **81.0** | **90.7** | **2.0** | **4.4** |

Table 10: Comparison of results with and without post-processing techniques on three datasets. † indicates the use of post-processing methods (Cheng et al., 2021; Bogolin et al., 2022).

| Method | R@1 ↑ | R@5 ↑ | R@10 ↑ | MdR ↓ | MnR ↓ |
|---|---|---|---|---|---|
| **MSR-VTT** | | | | | |
| Ours (MBDA) | 52.4 | 75.9 | 85.4 | 1.0 | 13.2 |
| Ours† (MBDA) | 55.3 | 78.2 | 87.7 | 1.0 | 10.3 |
| **DiDeMo** | | | | | |
| Ours (MBDA) | 53.1 | 78.7 | 86.5 | 1.0 | 10.4 |
| Ours† (MBDA) | 57.9 | 82.1 | 88.0 | 1.0 | 9.3 |
| **ActivityNet** | | | | | |
| Ours (MBDA) | 49.6 | 79.4 | 89.3 | 2.0 | 5.2 |
| Ours† (MBDA) | 57.9 | 83.9 | 92.0 | 1.0 | 4.3 |

## A.5 QUANTITATIVE RESULTS

**Similarity between video and text**   We calculate the average similarity between the decoupled components and the text embeddings on multiple test sets (using the component that is more similar to the text for each sample). As shown in the table 11, the average similarity scores are consistently higher than those of the baseline. For example, on the MSR-VTT dataset, our method achieves an average similarity of approximately 0.293, significantly outperforming the baseline score of 0.197. These results demonstrate that our method successfully preserves key information relevant to the text, while reducing redundant information.

**Cases**   We first conducted a case study by running inference using the trained model on examples from MSR-VTT. As table 12 shows, we identified examples with clearly dominant temporal or spatial information and measured the similarity between each of the decoupled components and the corresponding text embedding.

**Similarity between decoupled components**   We compute the cosine similarity between the two decoupled components, $V_{\text{spatial}}$ and $V_{\text{temporal}}$, on the test sets of several datasets, as shown in the table 13. For example, on the MSR-VTT test set, the average cosine similarity is approximately 0.052. Furthermore, after projecting both components into the frequency domain using the Discrete Fourier Transform (DFT), their average cosine similarity further drops to 0.0042. Both values are close to zero, indicating that our decoupling is orthogonal not only in theory but also quantitatively.

Table 11: Cosine Similarity (Avg.) on Different Datasets

| | cosine similarity (Avg.) | | |
|---|---|---|---|
| | MSR-VTT | DiDeMo | MSVD |
| ours | **0.293** | **0.315** | **0.288** |
| baseline | 0.197 | 0.203 | 0.149 |

Table 12: Case Study: Similarity Scores for Temporal and Spatial Examples

| Dominant Information | Example Text | Temporal Sim. | Spatial Sim. |
|---|---|---|---|
| Temporal | "a little girl does gymnastics" | **0.364** | 0.257 |
| | "men are doing wrestling" | **0.311** | 0.237 |
| Spatial | "cartoon show for kids" | 0.175 | **0.373** |
| | "a man with a very red nose" | 0.194 | **0.312** |

Table 13: Cosine Similarity Between Decoupled Spatial and Temporal Components.

| Dataset | MSR-VTT | DiDeMo | MSVD |
|---|---|---|---|
| Cosine similarity | 0.052 | 0.064 | 0.059 |
| Cosine similarity (After DFT) | 0.0042 | 0.0076 | 0.0093 |

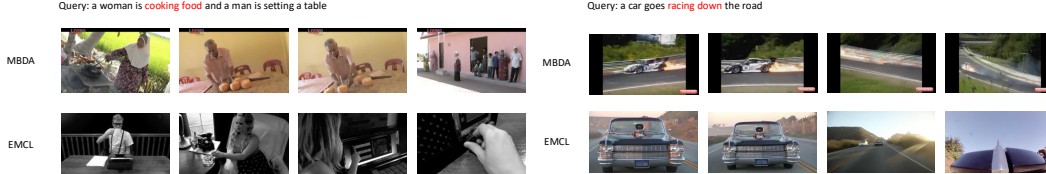

Figure 5: Qualitative results of text-video retrieval on MSR-VTT. Given a text query, we present the correct matched video returned by MBDA in the first row, and show the false result of EMCL (Jin et al., 2022) in the second row. The word highlighted in red indicates the key content (about motion) missed in the false result.

## A.6 USE OF LLMs

We use LLMs to polish writing. Moreover, we use LLMs to generate textual data on four datasets for experimental need, the detailed information is written in the implementation settings of Appendix A.1.

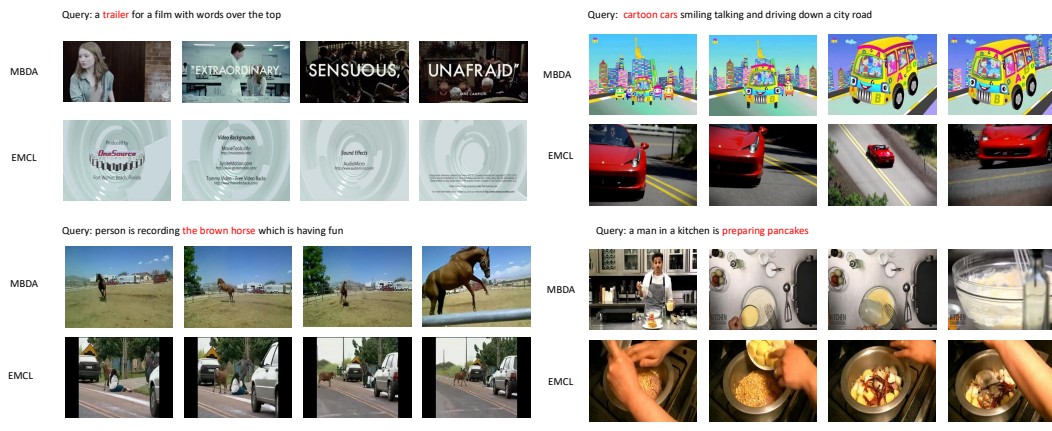

Figure 6: Qualitative results of text-video retrieval on MSR-VTT. Given a text query, we present the correct matched video returned by MBDA in the first row, and show the false result of EMCL (Jin et al., 2022) in the second row. The word highlighted in red indicates the key content (about details) missed in the false result.

