# OpenReview forum: "Modality-Balanced Decoupling Alignment for Text-Video Retrieval"
_ICLR.cc/2026/Conference — ICLR 2026 Conference Withdrawn Submission_

### Official Review · Reviewer_SNqC · 2025-10-27

**Soundness:** 3
**Presentation:** 2
**Contribution:** 2
**Rating:** 6
**Confidence:** 3

**Summary:**

The paper presents Modality-Balanced Decoupling Alignment (MBDA), a method for improving text-video retrieval tasks by addressing the challenges caused by the imbalance in information capacity between video and text. The key innovation is decoupling the video representations into orthogonal components, facilitating a more balanced alignment between video and text embeddings. The method includes two primary modules: Modality Proximity Alignment (MPA): This module aligns video and text representations closer in feature space. Video Representation Orthogonal Decoupling (VROD): It decouples the video features into two orthogonal components, enabling better alignment with text features. The MBDA method is validated through extensive experiments, showing significant performance improvements on benchmark datasets like MSR-VTT, DiDeMo, MSVD, and ActivityNet, outperforming existing state-of-the-art methods.

**Strengths:**

**Interesting Solution:** The decoupling of video features into orthogonal components addresses the semantic imbalance and high coupling within video data.

**Strong Experimental Results:** MBDA outperforms several baseline and state-of-the-art methods on multiple benchmark datasets, showing its practical effectiveness.

**Weaknesses:**

**Complexity**: The orthogonal decoupling and alignment process may introduce computational overhead, particularly during the decoupling stage.

**Questions:**

Could the authors compare the video feature encoding time for baseline and your proposed method? Not the total retrieval time, just the visual feature encoding time.

---

### Official Review · Reviewer_sfyV · 2025-10-31

**Soundness:** 3
**Presentation:** 4
**Contribution:** 2
**Rating:** 4
**Confidence:** 4

**Summary:**

This paper presents MBDA, a method for text–video retrieval that aims to reduce the semantic gap caused by the information imbalance between visual and textual modalities.
MBDA includes two main components:1) MPA (Modality Projection Alignment), which aligns video and text embeddings in a shared space.2) VROD (Video Representation Orthogonal Decomposition), which divides the video embedding into two orthogonal parts to balance its information capacity with the corresponding text.
The decoupling is achieved through a low-rank decomposition process and analyzed in the frequency domain using the Discrete Fourier Transform (DFT).
Extensive experiments on four public benchmarks demonstrate that MBDA delivers competitive performance and validates the effectiveness of its design.

**Strengths:**

S1. The paper identifies two key challenges in text–video retrieval: the imbalance in information capacity between modalities and the strong coupling of video information. These insights motivate the proposed method effectively.

S2.Comprehensive experiments are conducted on four benchmarks, supported by detailed ablation studies that confirm the effectiveness of each module.

**Weaknesses:**

W1. Although the DFT proofs claim orthogonality between decomposed components, no quantitative metrics are provided to show that feature redundancy is actually reduced.

W2. The paper argues that orthogonality leads to balanced text–video representations, but text features remain monolithic. There is no evidence that decomposed video features align meaningfully with text semantics.

W3. The paper states that video representations become comparable in scale to text after decomposition, but this is not supported by evidence. The t-SNE visualization still shows that video features are much more spread out.

W4. The method relies on text–video pairs for fine-tuning. It does not support zero-shot or few-shot retrieval, limiting its scalability and adaptability.

**Questions:**

refer to weakness

---

### Official Review · Reviewer_qpw7 · 2025-10-31

**Soundness:** 2
**Presentation:** 2
**Contribution:** 2
**Rating:** 4
**Confidence:** 4

**Summary:**

The paper introduces MBDA, a novel framework for text-video retrieval designed to solve two core issues: the semantic capacity imbalance between rich video features and coarse text features, and the highly entangled nature of video information. MBDA consists of a MPA module to pull distributions closer, and a VROD module to split video features into orthogonal components (spatial and temporal). The proposed method achieves state-of-the-art results on four benchmarks.

**Strengths:**

1. MBDA achieves state-of-the-art performance across four public benchmarks.
2. The proposed method is new to me.

**Weaknesses:**

1. The information gap between video and text has been investigated in the past few years, see [1][2]. A more detailed comparison or clear credit to those works is necessary.
2. Also, a clear definition of the information gap is needed. Besides, how the information gap is addressed with the proposed method is also important. Some quantitative metrics or results are needed.
3. Though the theoretical proof shows that spatial and temporal features are orthogonal, it is still remains unknown how empirically they are orthogonal (Eqs. 4 and 5).
4. Writing can be further improved. Lots of the notations come without explanations.

[1] Multiple query video retrieval. ECCV 2022.
[2] DREAM: Improving Video-Text Retrieval Through Relevance-Based Augmentation Using Large Foundation Models. NAACL 2025.

**Questions:**

1. Line 212. How to obtain H(V)? What if the video comes without tags, titles, and summaries?
	1. Also, as evidenced by the results in table 2, VROD only is worse than MPA only.
2. Moreover, as in some benchmark datasets, such as MSR-VTT, their text query comes from the rewritten captions/titles, this might lead to severe label leakage.
3. See weakness.

---

### Official Review · Reviewer_mHTm · 2025-11-01

**Soundness:** 2
**Presentation:** 3
**Contribution:** 3
**Rating:** 4
**Confidence:** 4

**Summary:**

This paper proposes MBDA to address modality imbalance in text-video retrieval by first employing an MPA module to bring video features closer to the text space, then utilizing a VROD module to decouple video features into two orthogonal temporal and spatial components, achieving more balanced cross-modal alignment and state-of-the-art performance across four benchmark datasets.

**Strengths:**

1. The experiments show significant performance improvements.2、

2. The orthogonal decoupling approach appears to be well-motivated.

3. The writing quality is good.

**Weaknesses:**

1. Incomplete similarity progression analysis. The paper should provide complete cosine similarity evolution: in original features, after MPA, and after VROD. Table 13 only reports post-decoupling similarity (0.052) without intermediate baselines, making it impossible to quantify each component's contribution.
2. I'm wondering whether orthogonality is learned through GLU modules or inherently enforced by complementary downsampling, since sparse sampling naturally decorrelates information. The authors should provide ablation with fixed downsampling (without GLU) to isolate learnable parameters' contribution. If fixed downsampling already achieves ~0.05 similarity, orthogonality primarily stems from architectural design rather than learned decomposition.
3. Would adding explicit orthogonality constraint loss improve performance? Currently the model relies solely on architectural implicit constraints without learnable optimization toward orthogonality. This experiment would reveal whether there's room for further orthogonalization beyond what the architecture provides.

**Questions:**

See the weakness

---

### Note · Authors · 2025-11-14

I have read and agree with the venue's withdrawal policy on behalf of myself and my co-authors.